
# The characteristics of atmospheric brown carbon in Xi'an, inland China: sources, size distributions and optical properties

Can Wu[1,2], Gehui Wang[1,2,3,4*], Jin Li[2], Jianjun Li[2], Cong Cao[2], Shuangshuang Ge[1], Yuning Xie[1], Jianmin Chen[3,5], Xingru Li[1,6], Guoyan Xue[1], Xinpei Wang[1], Zhuyu Zhao[7], Fang Cao[7]

[1]Key Lab of Geographic Information Science of the Ministry of Education, School of Geographic Sciences, East China Normal University, Shanghai 210062, China
[2]Key Lab of Aerosol Physics and Chemistry, State Key Laboratory of Loess and Quaternary Geology, Institute of Earth Environment, Chinese Academy of Sciences, Xi'an 710061, China
[3]Institute of Eco-Chongming, 3663 North Zhongshan Road, Shanghai 200062, China
[4]CAS Center for Excellence in Regional Atmospheric Environment, Institute of Urban Environment, Chinese Academy of Sciences, Xiamen 361021, China
[5]Department of Environmental Science and Technology, Fudan University, Shanghai 200433, China
[6]Department of Chemistry, Analytical and Testing Center, Capital Normal University, Beijing 100048, China
[7]Yale-NUIST Center on Atmospheric Environment, Nanjing University of Information Science & Technology, Nanjing 210044, China

*Corresponding author. Prof. Gehui Wang
E-mail address: ghwang@geo.ecnu.edu.cn, or wanggh@ieecas.cn (Gehui Wang)



**Abstract**: To investigate the characteristic of atmospheric brown carbon (BrC) in the
semi-arid region of East Asia, $PM_{2.5}$ and size-resolved particles in the urban atmosphere of
Xi'an, inland China during the winter and summer of 2017 were collected and analyzed for
optical properties and chemical compositions. Methanol extracts (MeOH-extracts) were more
light-absorbing than water extracts ($H_2O$- extracts) in the optical wavelength of 300-600 nm,
and well correlated with nitrophenols, polycyclic aromatic hydrocarbons (PAHs) and
oxygenated PAHs ($R^2 > 0.6$). The light absorptions ($abs_{\lambda=365nm}$) of $H_2O$- extracts and
MeOH-extracts in winter were 28±16 M/m and 49±32 M/m, respectively, which are about 10
times higher than those in summer, mainly due to the enhanced emissions from biomass
burning for house heating. Water extracted BrC predominately occurred in the fine mode (<
2.1 µm) during winter and summer, accounting for 81% and 65% of the total absorption of
BrC, respectively. The light absorption and stable carbon isotope composition measurements
showed an increasing ratio of $abs_{\lambda=365nm}$-MeOH to $abs_{\lambda=550nm}$-EC along with an enrichment of
$^{13}C$ in $PM_{2.5}$ during the haze development, indicating an accumulation of secondarily formed
BrC (e.g., nitrophenols) in aerosol aging process. PMF analysis showed that biomass burning,
fossil fuel combustion, secondary formation, and fugitive dust are the major sources of BrC in
the city, accounting for 54.7%, 19%, 16.2%, and 10% of the total BrC of $PM_{2.5}$, respectively.

**Key words:** Brown Carbon; Haze; Stable carbon isotope composition; Biomass burning;
Secondary formation.




## 1. Introduction

Brown carbon (BrC) is a small fraction of carbonaceous aerosols, but it exhibits strong

absorption abilities from near ultraviolet (UV) to visible light regions, and thus has been given

extensive investigation in the recent decades (Laskin et al., 2015;Yan et al., 2018;Gustafsson

et al., 2009). BrC has significant impact on climate change directly by absorbing solar

radiation and indirectly by accelerating snowmelt and affecting the albedo (Qian et al.,

2015;Andreae and Ramanathan, 2013). Based on the remote sensing observations and

chemical transport models (Chung et al., 2012;Wang et al., 2014;Jo et al., 2016), a

non-negligible positive radiative forcing by BrC was found on a global scale with a range

from 0.1 to 0.6 W m$^{-2}$. Beyond that, BrC also influence the atmospheric chemistry and human

health. For example, BrC can shield polycyclic aromatic hydrocarbons (PAHs) from being

oxidized, and thus substantially elevate lung cancer risk from PAHs (Hsu et al., 2014;Yan et

al., 2018).

The sources of BrC are complicated, which can be primarily emitted from incomplete

combustion of carbon-containing materials (e.g., biomass, coal and petroleum products.) and

secondarily derived from aqueous-phase reaction (Sun et al., 2017;Gilardoni et al., 2016;Xie

et al., 2018;Nakayama et al., 2013). Biomass burning was found to be a major primary source

of BrC (Chen and Bond, 2010;Chakrabarty et al., 2010;Saleh et al., 2014), because lignin is of

an unsaturated benzene-like structure, which is a chromorphore group. Field measurements

and laboratory studies found that BrC is also of secondary sources by forming chromophores

during the atmosphere ageing process, e.g., high-NOx photooxidation (Liu et al., 2016;Xie et

al., 2017), ozonolysis of aromatic precursors (Lee et al., 2014), and aqueous-phase



photochemical oxidation and polymerization (Smith et al., 2014;Flores et al., 2014;Bones et
al., 2010). BrC products account for very small weight fraction of organic aerosol (OA), but
have a significant effect on OA optical properties. For example, nitroaromatic compounds that
are generated by photooxidation of toluene under high NOx conditions may account for
40-60% of the total light absorption of toluene-SOA (Lin et al., 2015).

Multiple approaches have been developed to quantify the light absorption properties of

BrC (Moosmuller et al., 2009), and a common and sensitive approach is the direct
measurement of spectrophotometric properties of aerosol water or filter extracts by using
optical instrumentation. The advantage of this method can avert interference from insoluble
absorption material (e.g., black carbon)(Cheng et al., 2016;Shen et al., 2017), and supply
high-resolution spectrum over a wide wavelength coverage. Furthermore, it is favorable for
characterization of BrC light-absorbing components by combing with other analytical
techniques, such as mass spectrometry (MS)(Laskin et al., 2015;Corr et al., 2012;Satish et al.,

2017).

Xi'an is a metropolitan city located in Guanzhong Basin of inland China, which is a

typical semiarid region in East Asia and have been suffering from serious particle pollution
due to the large emission of anthropogenic pollutants (Wu et al., 2018;Wang et al., 2016;Wu
et al., 2019), especially intensive coal combustion and biomass burning in winter for house
heating (Wang et al., 2017). Many studies have been conducted on the BrC optical properties
in China, but most of which were based on $PM_{2.5}$ and $PM_{10}$ sample collection and focused on
the bulk aerosol optical properties with no information on the size distributions (Shen et al.,
2017;Huang et al., 2018). In this study, both $PM_{2.5}$ and size-segregated aerosol samples in



Xi'an were collected during the 2017 winter and summer and analyzed for the characteristics
of BrC. We firstly investigated the seasonal variations of chemical composition and
light-absorption of BrC in the city, then discussed the size distribution of BrC and the impact
of aerosol ageing process on BrC, and finally quantified its source contributions.
**2. Experimental section**
**2.1 Sample collection**

Aerosol samples with a 12-hr interval were collected using a high-volume (~1.13 m$^3$

min$^{-1}$) air sampler (Tisch Environmental, Inc., OH, USA) from December 31, 2016 to January
22, 2017 (in winter) and from July 18 to August 6, 2017 (in summer). The sampler was
installed on the roof of a three-story building on the campus of the Institute of Earth
Environment, CAS (34.22ºN, 108.88ºE), which was located at the urban center of Xi´an,
inland China. Meanwhile, size-resolved aerosols with 9 size bins (cutoff points were 0.43,
0.65, 1.1, 2.1, 3.3, 4.7, 5.8, and 9.0 µm, respectively) were collected by using an Anderson
sampler at an airflow rate of 28.3 L min$^{-1}$ for 24 hr. All samples were collected onto the
pre-baked (450℃ for 6 hr) quartz filters and stored in a freezer (-18℃) prior to analysis.
**2.2 Chemical analysis**

A punch (0.526 cm$^3$) from each PM$_{2.5}$ filter sample was analyzed for organic carbon (OC)

and elemental carbon (EC) with a DRI Model 2001 Thermal/Optical Carbon Analyzer
(Atmoslytic Inc., Calabasas, CA, USA) following the IMPROVE-A protocol (Chow et al.,
2007). More details of the method including quality assurance and quality control (QA/QC)
can be found elsewhere (Wang et al., 2010).

Partial filters were cut into pieces, and then extracted three times under sonication with





15ml Milli-Q pure water (18.2 MΩ). Ten ions such as $SO_4^{2-}$, $NO_3^-$, $Cl^-$, $NH_4^+$, and $K^+$ were
determined using ion chromatography (Dionex, ICS-1100). Similar extraction processes were
also applied to measure the water-soluble organic carbons (WSOC) of the samples by
following the method of Wang et al. (2013). In order to analyze the organic compounds in the
samples such as levoglucosan, PAHs, OPAH and nitrophenols, aliquot of the filter was
extracted with a mixture of methanol and DCM (1:5, v/v), derivatized with BSTFA and
measured by using gas chromatography (HP 7890A, Agilent Co., USA) coupled with mass
spectroscopy detector (GC/MS) (HP 5975, Agilent Co., USA). Details of sample extraction
and derivatization were documented elsewhere (Wang et al., 2009b;Ren et al., 2017). Stable
carbon isotope composition of total carbon ($\delta^{13}C_{TC}$) was determined by using an elemental
analyzer (EA) (Carlo Erba, NA 1500) coupled with an isotope ratio mass spectrometer (IRMS,
Finnigan MAT Delta Plus), more details of the method can be referred to elsewhere (Cao et al.,

2016).

**2.3 Light absorption measurements**
Brown carbon (BrC) was extracted from a size of 6 cm$^3$ filter samples for 30min
ultrasonication with 20ml Milli-Q pure water or methanol. All extracts were then filtered
through 0.45 μm PTFE (for water) and 0.22 μm PES (for methanol) pore syringe filter to
remove insoluble components and filter remnants. The light-absorption spectra were analyzed
with a UV–visible spectrophotometer (AOE INSTRUMENTS, China) over a wavelength
range of 190–900 nm (Hecobian et al., 2010). The absorption coefficient of water or methanol
extracts (M m$^{-1}$) could be calculated as the following equation (Teich et al., 2017):



$$abs_\lambda = (A_\lambda - A_{700})\frac{V_1}{V_a \times L} \times \ln(10) \qquad (1)$$

Where $A_\lambda$ and $A_{700}$ were the light absorption of the extracts at the wavelength of $\lambda$ and
700nm, respectively. $V_1$ represented the volume of the solvent extracting the filter sample, and
$V_a$ was referred to the volume of air corresponding to the filter punch. L was the absorbing
path length (i.e., 1 cm for the currently used quartz cuvettes). The ln(10) was converted from
base 10 (the form provided by the spectrophotometer) to natural logarithms. According to the
previous studies, the absorption coefficient at 365nm was used as the brown carbon
absorption in order to avoid disturbance of inorganic salts such as nitrate.
The bulk mass absorption coefficient (MAC, $m^2$/g) of the extracts at a given wavelength
can be described by the following equation:
$$MAC = \frac{abs_\lambda}{C_{W(M)SOC}} \qquad (2)$$

Where $C_{WS(OC)}$ was the WSOC mass concentration of the water extracts or
methanol-soluble organic carbon (MSOC) mass concentration of the methanol extracts. In this
study, we assumed that OC could be completely dissolved in methanol solvent and substituted
the MSOC to participate in the calculation. This hypothesis would lead to somewhat
underestimation of the MAC of the methanol extracts, although high extraction efficiency of
methanol solvent had reported by previous studies (Liu et al., 2013) .
The wavelength dependence of light-absorption with respect to the empirically defined
power law relationship described by the following equation (Laskin et al., 2015):
$$MAC = K\lambda^{-AAE} \qquad (3)$$





Where K is a factor that includes aerosol mass concentrations, the AAE is termed as
absorption Angström exponent. In this study, the AAE value of the filter extracts was
determined by a linear regression of $\log(abs\lambda)$ versus $\log(\lambda)$ over a wavelength range of
300-450nm.

**2.4 Positive Matrix Factorization (PMF) source apportionment**

PMF, as a receptor model, decomposes the sample matrix into two matrices (factor
contributions and factor profiles), and has been widely used in source apportionment of
atmospheric pollutants. More details on PMF can be found on the EPA website
(https://www.epa.gov/air-research/epa-positive-matrix-factorization-50-fundamentals-and-use
r-guide). In the present work, the mass concentrations of major species (OC, EC, WSOC,
$SO_4^{2-}$, $NO_3^-$, $NH_4^+$, $Ca^{2+}$), organic markers (benzo(b)fluoranthene (BbF), benzo(e)pyrene
(BeP),indeno(1,2,3-c,d)pyene (IP), levoglucosan, and nitrophenols), and $abs_\lambda$ of water extracts
have been used as the input data to perform the source apportionment for brown carbon with
the EPA PMF 5.0 version, similar reports have been found elsewhere (Hecobian et al., 2010).
The model was run numerous times with 3–7 factors and various combinations of the
concentration and absorption data set. Base on the Q value (Q $_{true}$ and Q $_{robust}$) and $r^2$, which
are indicative of the agreement of the model fit, four factors were obtained as the optimal
solution.

**3. Results and discussion**

**3.1 Carbonaceous species in PM$_{2.5}$ during summer and winter**

Figure 1 shows the temporal variations in the concentrations of PM$_{2.5}$, WSOC, OC and
$abs_{\lambda=365nm}$ value during the two seasons. WSOC varied from 5.3 to 67 μg/m$^3$ in winter with an



average of 23 ± 13 μg/m$^3$ (Table 1), which was 4.0 times higher than that in summer. OC
exhibited a similar seasonal variation with WSOC with an average of 41 ± 25 μg/m$^3$ in winter
and 8.4± 2.4 μg/m$^3$ in summer, respectively. Whereas, WSOC/OC ratio was much higher in
summer (0.70 ± 0.12) than that in winter (0.58 ± 0.13), partly as a result of an enhanced
photochemical formation of WSOC under the intense sunlight conditions, similar phenomena
were also found in Beijing (Ping et al., 2017), Shanghai (Zhao et al., 2015a), Tokyo (Miyazaki
et al., 2006) and Southeastern United States (Ding et al., 2008).

PAHs, OPAHs, and nitrophenols, as ubiquitous matters in the atmosphere, are mainly

derived from combustion emission (e.g., coal, biomass) (Wang et al., 2015), although OPAHs
and nitrophenols can also be derived from secondary formation (Lin et al., 2015;Xie et al.,
2017). These matters are the efficient light-absorbing compounds due to owning specific
light-absorbing molecular structures-BrC chromophores (Lin et al., 2017;Bluvshtein et al.,
2017). Herein, 14 PAHs, 7 OPAHs, and 7 nitrophenols were examined for investigating their
effect on BrC absorption. As seen in Figure S1, the temporal variations of PAHs, OPAHs, and
nitrophenols were similar with levoglucosan, which is recognized as the tracer of biomass
burning emissions, indicating that biomass burning is one of the major sources of these
compounds .Concentrations of PAHs, OPAHs, and nitrophenols during winter were 149 ± 89
ng/m$^3$, 174 ± 98 ng/m$^3$ and 17 ± 12 ng/m$^3$ (Table 1), respectively, and were 10 - 43 times
higher than those in summer, which can be explained by an increasing emission from
residential heating during winter in the city and its surrounding regions.

As shown in Table S1, abs$_{\lambda=365nm}$ extracted by methanol displays well correlations with

PAHs, OPAHs, and nitrophenols, especially in winter (R$^2$ > 0.80), which suggests that those





species are important light absorption contributors for BrC in Xi'an. Huang et al. (2018)
found that PAHs and OPAHs in Xi'an accounted for, on average, 1.7% of the overall
absorption of methanol-soluble BrC, but their mass fraction in OC was only 0.35%. A recent
study reported that biomass burning also emitted nitroaromatic compounds, particularly
nitrophenols, and accounted for 50-80% of the total visible light absorption (> 400 nm) (Lin
et al., 2017). The robust correlations of above compounds with the absorption at $\lambda$=365 nm
suggest that PAHs, OPAHs and nitrophenol are strong light-absorbing species.
**3.2 Light absorption of BrC in water and methanol extracts**
**3.2.1 Seasonal variations of light absorption by BrC**
As shown in Figure 2a and 2b, the marked feature of BrC in Xi'an is that the absorption
spectrum increased notably from the visible to the ultraviolet ranges, and the average
abs-MeOH at $\lambda$=365 nm was 1.5 - 1.7 times higher than abs-$H_2O$ in the two seasons,
indicating that MSOC provided a more comprehensive estimation for BrC. Due to enhanced
emission of BrC, average $abs_{\lambda=365nm}$ of BrC found in winter was 49 ± 32 M/m for MeOH and
28 ± 16 M/m for WSOC, which were 9.5- and 8.1-fold higher than that in summer. This
phenomenon was also observed in previous studies in Xi'an (Shen et al., 2017;Huang et al.,
2018) and other areas of China (Du et al., 2014;Chen et al., 2018). Compared with other
regions (Table 2), the absolute $abs_{\lambda=365nm}$ values in Xi'an were slightly lower than that in
Indo-Gangetic Plain, India (Satish et al., 2017;Bachi, 2016), but were considerably higher
than that in Beijing, China (Du et al., 2014), US (Zhang et al., 2011) and Korea (Kim et al.,
2016), suggesting that BrC pollution is more significant in Xi'an, a developing region in
China. Furthermore, enhanced $abs_{\lambda=365nm}$ loading in the nighttime was observed during the



two seasons, which can be ascribed to the shallower boundary layer height and the absence of
photo-bleaching processes in nighttime (Saleh et al., 2013;Zhao et al., 2015b).

Linear regression slopes on the scatter plots of $abs_{\lambda=365nm}$ values versus WSOC or OC

represented the average of MAC at 365 nm (i.e., $MAC_{WSOC}$ and $MAC_{MSOC}$). During winter,
there was a slight disparity between the $MAC_{WSOC}$ and $MAC_{MSOC}$ with the averages of 1.2 ±
0.06 and 1.3 ± 0.03 $m^2$/g (Figure 2e), respectively, which indicates that there are some similar
chromophores of BrC between the two fractions. $abs_{\lambda=365nm}$ showed a strong linear correlation
with levoglucosan ($R^2 > 0.97$), suggesting that abundant BrC may be largely derived from
biomass burning. As shown in Fig. S2, mass ratios of levoglucosan/mannosan and
levoglucosan/galacosan in the $PM_{2.5}$ samples are similar to biomass types (i.e., woods, leaves,
wheat straw), again reflecting that biomass burning combustion in Xi'an and its surrounding
regions are probably the major sources of BrC in the city during winter. Compared to winter,
the MAC in summer was slightly lower, which can be in part attributed to the less abundant
light-absorbing PAHs and OPAHs due to no biomass burning for house heating. Moreover,
with increasing photooxidation in summer, fragmentation reactions would occur and thus
decrease light absorption for BrC aerosols, as reported by Sumlin et al. (2017), because higher
levels of $O_3$ and OH radicals in summer intensify the photooxidation and diminish the BrC
aerosol light absorption by reducing the size of conjugated molecular systems. Interestingly,
we found that the $MAC_{WSOC}$ (1.1 ± 0.2 $m^2$/g) in summer was significantly enhanced compared
to $MAC_{MSOC}$ (0.8 ± 0.1 $m^2$/g), which can be ascribed to more amount of non-BrC in methanol
extracts. The $abs_{\lambda=365nm}$ showed a poor correlation with levoglucosan (Table S1), further
indicating that the biomass burning was not the dominant source for BrC in summer.



Absorption Ångström exponents (AAE), which were derived from the filter methanol-
and water-extracted BrC ($AAE_{WSOC}$ and $AAE_{MSOC}$) for wavelengths between 300 and 450 nm,
were $6.1 \pm 9.7$ and $5.3 \pm 8.5$ (Table 2) in winter, respectively, and resembled that in Beijing
(Cheng et al., 2016), Guangzhou (Liu et al., 2018) and Indo- Gangetic Plain (Bachi, 2016),
possibly indicating that the chemical compositions of BrC chromophores in these regions are
similar during winter. As seen in Table 2, unlike those of $H_2O$-extracts, the averaged values of
MAC and AAE of MeOH extracts were 40% and 10% higher in winter than in summer,
suggesting that chemical compositions of BrC are different between the two seasons in the
city and the winter BrC contained more non-polar compounds that are of stronger
light-absorbing ability.
**3.2.2 Aerosol size distribution of BrC**
Particles with different sizes are of different chemical compositions, and thus optical
properties of BrC in different size of particles are also different (Zhang et al., 2015;Zhai et al.,
2017). However, information on size distribution of BrC absorption is very limited. In this
study, we mainly focused on the water-extracted samples, because particles deposited on the
filter surface are unevenly distributed, making the quantifications of OC and EC in the
size-segregated samples not accurate enough. As shown in Figure S3, there was a good
relationship between the $abs_{\lambda=365nm}$ ($R^2 > 0.96$) of the samples collected by Anderson sampler
and those collected by high-volume $PM_{2.5}$ sampler (Fig. S3), suggesting a good agreement
between the two sampling methods.
As show in Figure 3, $abs_{\lambda=365nm}$ presented a bimodal pattern during winter and summer,
dominating at the fine mode (Dp <2.1μm) with relative contributions of 81% and 65% to the



total absorption in the two seasons, respectively. These proportions were similar to those
reported for a forest wildfire event, which showed that 93% of the total BrC absorption was in
the fine particles (0.10 < Dp < 1.0 µm) (Lorenzo et al., 2018). Maximum absorptions were
observed at 1.02 and 0.71µm (Dpg- geometric mean diameters, Figure 3a and 3b) in winter
and summer, respectively, which is in agreement with the observations by Lei et al (2018),
who found that the major peaks for BrC absorption were in the rang from 0.5µm to 1.0µm in
urban and may shift toward smaller size (< 0.4µm) for particles released from burning
experiments (Lei et al., 2018). However, the size distribution pattern of MAC was different
from that of $abs_{\lambda=365nm}$ in Xi'an, which presented a monomodal distribution with a peak in the
fine mode (<2.1µm) in winter and a bimodal distribution in summer with two peaks in the fine
(<2.1µm) and coarse (>2.1µm) modes, respectively (Figure 3c and 3d). As seen in Figure 3c
and 3d, the fine mode of MAC was around 50% larger in winter than that in summer,
suggesting that water-soluble fraction of winter fine particles was more light-absorbing
compared to that in summer, probably due to the summertime stronger bleaching effect.
**3.3 Underestimation of BrC absorption by solvent extraction methods**

A few studies pointed out that absorption properties of BrC extracted by bulk solution

may not entirely reflect the light absorption by ambient aerosols. Here, we further calculated
the light absorption of the samples using the Mie theory combined with an imaginary (*k*,
responsible for absorption) refractive index with assumptions that particles were spherical
morphology and externally mixed with other light-absorbing components. The imaginary
refractive index could be obtained from MAC using follow equation (Laskin et al., 2015):


$$k_{(\lambda)} = \frac{\rho\,\lambda\,abs}{4\pi \times WSOC} = \frac{\rho\,\lambda\,MAC}{4\pi} \qquad (4)$$


Where $\rho$ (g/cm$^3$) was particle density and assigned as 1.5, more details about Mie calculations
can be referred to the study by Liu et al. (2013).

As noted above, most BrC aerosols were in the fine mode (<2.1μm), thus, here we only

focused on this fraction for the Mie calculations. The values of imaginary refractive in winter
remains nearly constant (0.038-0.048) for different particle sizes at λ=365 nm (Table 3),
which was about two times smaller than that (0.093 ± 0.049) over Gangetic Plain, India
(Shamjad et al., 2017). Values of $k$ in summer were slight smaller when compared to those in
winter, suggesting that the aerosols in summer were more aged. Sumlin et al. (2017) found
that $k$ decreases with the atmospheric aging from 0.029 ± 0.001 to 0.019 ± 0.001 at λ=375 nm.
However, $k$ values in this study were 1.8 to 8.1 times higher than previously reported values
from the United States (Liu et al., 2013;Washenfelder et al., 2015). This is because that PM$_{2.5}$
particles in Xi'an, China are enriched in BrC and the mass absorption coefficient was
considerably higher than that in US. Figure 4 compares the difference between abs$_{\lambda=365nm}$
predicted by Mie theory and that extracted by the bulk solution. Mie theory predicted
abs$_{\lambda=365nm}$ was 1.3-fold higher than that measured by the bulk solution, suggesting that the
solvent extraction methods, which have commonly been used for atmospheric BrC
measurements, could result in a significant underestimation on optical absorption of aerosols.
**3.4 The characteristic of BrC with the aerosol aging**

During the ageing process secondary organic aerosols (SOA) with strong chromophores

can be generated and efficiently absorb solar radiation (Lin et al., 2014;Lin et al., 2016). From



Figure 5, it can be found that air quality in Xi'an during the winter varied from the clean
($PM_{2.5} < 75$ μg/m$^3$) to the polluted conditions ($PM_{2.5} > 75$ μg/m$^3$) from the period of 12$^{th}$
January to 19$^{th}$ January. Such a case provides an opportunity to investigate the changes in
light-absorption by BrC during the aerosol ageing process.

As shown in Figure 5a and 5b, $abs_{\lambda=365nm}$ extracted by water and MeOH in Xi'an during

the campaign showed an increasing trend from 12$^{th}$ January to 19$^{th}$ January, which is similar
to $PM_{2.5}$ loadings but opposite to the visibility, indicating that BrC is one of the important
factors leading to the visibility deterioration. From Figure 5b, it can also be seen that light
absorption of water-extracts dominated over the total BrC absorption especially in daytime
and showed a variation pattern similar to the $PM_{2.5}$ (Figure 5a) and WSOC loadings (Figure
5c), indicating a continuous formation of secondary BrC during the aerosol ageing process. To
illustrate this point, the stable carbon isotopic composition ($\delta^{13}C_{TC}$) of total carbon (TC) in the
samples was measured. WSOC/OC showed a positive correlation with the $\delta^{13}C_{TC}$,
demonstrating an ageing process of aerosols during the haze development from 12$^{th}$ to 19$^{th}$,
January, although it was weak (r = 0.47, n = 17). Similar conclusions were also reported by
Yang et al. (2004) and Pavuluri et al. (2015). From Figure 5c, increasing trends of OPAHs and
nitrophenols were observed during the haze development, suggesting that more SOAs with
chromophores were generated during such an aerosol ageing process, because these
compounds are also of secondary origins. To exclude the possible impact of the changes in
BrC source emissions, the values of PAHs/OC and levoglucosan/OC were applied in this study,
because PAHs and levoglucosan emission factors are different for different
sources(Nguyen-Duy and Chang, 2017). As shown in Fig. S4, both of values indistinctively





change during the aerosol ageing process, indicating that the increasing $abs_{\lambda=365nm}$ are not
caused by the changes in source emissions. Moreover, we found that $MAC_{MSOC}$ values during
the age process also increased (Figure 5a), further suggesting that the bleaching effect on
light-absorbing BrC was reducing during the haze developing process.

EC, which is also called as black carbon, is one of the major absorbing aerosol

components in the atmosphere (Collier et al., 2018;Peng et al., 2016). To further assess the
relative contribution of BrC during the aerosol ageing process, we compared the mass
absorption efficiency of EC at $\lambda$=550 nm (7.5 ± 1.2 m$^2$/g) with BrC by using the method
reported from Yan et al. (2015) and Kirillova et al. (2014). As shown in Figure 5c, the
concentrations of EC have a slight change in the haze period, so the changes in light
absorption of EC remained nearly constant. However, the ratio of
$abs_{\lambda=365nm}$-MeOH/$abs_{\lambda=550nm}$-EC increasingly became larger along with the visibility
deterioration from January 12$^{th}$ to January 19$^{th}$ (Fig. 5b), while the mass ratios of PAHs/EC,
OPAHs/EC and nitrophenols /EC during the period showed a significant negative correlation
with visibility (Fig. S5), further suggesting that the impact of BrC on the visibility was more
significant in comparison with EC.

During the haze developing process organic aerosols are usually getting more aged and

enriched in heavier $^{13}$C due to the kinetic isotopic effect (KIE) (Wang et al., 2010). As shown
in Figure 6a and b, $\delta^{13}$C of PM$_{2.5}$ samples presented a strong positive correlation with $abs_{\lambda=365}$
$_{nm}$-MeOH (R$^2$=0.68) in the daytime, while there was no such a correlation in the nighttime
during the haze period of January 12$^{th}$ -19$^{th}$, indicating a daytime formation of secondary BrC.
From Figure 6c and 6d, we also found that the correlation of $abs_{\lambda=365 nm}$-MeOH/ $abs_{\lambda=550 nm}$-EC



ratio with nitrophenol was much stronger in daytime than in nighttime, which is opposite to
the correlation of $abs_{\lambda=365\ nm}$-MeOH/ $abs_{\lambda=550\ nm}$-EC ratio with PAHs. Nitrophenols can be
produced from secondary photooxidation of phenol with NOx, while PAHs are produced
solely from direct emissions especially from coal and biomass burning for house heating. The
opposite diurnal correlations of $abs_{\lambda=365\ nm}$-MeOH/ $abs_{\lambda=550\ nm}$-EC ratio with nitrophenols and
PAHs again revealed an enhanced formation of secondary BrC during the aerosol ageing
process.

**3.5 Positive matrix factorization (PMF) analysis for BrC source apportionment**

In the current work, The EPA PMF 5.0 model was used for identifying the possible
sources of BrC. Because the number of the collected samples in each season was not large
enough, data from the two seasons were merged together to form a dataset of 80 × 12 (80
samples with 12 species) in order to obtain an accurate analysis according to the PMF user
guide. The resolved source profiles (factors) represented the sources that influenced
variability in the selected components throughout two seasons in Xi'an. Similar approach was
also reported by Zhang et al. (2010). With several iterative testes, a solution with four factors
was identified as the optimal solution. As shown in Table S2, the values of $Q_{true}$ and $Q_{robust}$
were consistent, which indicates that the model fits the input data well. Furthermore, the
correlation coefficient between input and model values ranged from 0.82 to 0.99 with an
average 0.96, also implying that the model fit well. This assess method was widely used in
previous studies (Ren et al., 2017;Wang et al., 2009a).
Figure7 shows the factor profiles resolved by the model. Factor 01 was characterized by
high levels of BeF (52.4%), BeP (56.5%), and IP (67.2%), which were primarily derived from



coal combustion and vehicle exhausts (Kong et al., 2010;Ma et al., 2010;Harrison et al., 1996),
further, relatively high OC (28.5%) and EC (25%) associated with this factor was well known
tracers of exhaust emissions (Zong et al., 2016), so we identified Factor 01as the source from
fossil fuel combustion. Factor 02 (fugitive dust) shows high contribution of $Ca^{2+}$ (66.8%) and
a moderate loading of EC (39.3%). Ca, as one of the most abundant crustal elements, is
largely from construction work, resuspended dust or soil sources (Chow et al., 2004;Han et al.,
2007). In addition, EC was a well-known tracer of vehicular emissions (Dorado et al., 2003),
so this factor can be attributed to the impact of vehicles passing with higher speeds, leading to
resuspend non-tailpipe particles. Moreover, the concentrations of $Ca^{2+}$ in the night were
almost higher than that during the day time, with averages of $1.8 \pm 1.56$ and $1.43 \pm 0.85$ μg/m$^3$,
respectively. This is consistent with time for transporting the construction wastes by lorry.
Thus, factor 02 was identified as fugitive dust. Factor 03 was identified as secondary
formation, as it is associated with high loadings of $NO_3^-$ (62.8%), $SO_4^{2-}$ (72.8%), $NH_4^+$ (68.8%)
and a moderate loading of OC and WSOC, indicating the presence of secondary inorganic and
organic aerosols. The factor 04 showed high loadings with nitrophenols, levoglucosan, and
abs-MeOH and was identified as biomass burning, because levoglucosan is the tracer for
biomass burning smoke, and nitrophenols can be produced in the aging process of biomass
burning plume.

Figure 8 shows the contributions of the above sources to the light absorption at λ=365nm,

which also represents the fraction of brown carbon for the factors. Biomass burning was the
primary source of the BrC, accounting for 54.7% of the total BrC in the city, which is
coincided with the results discussed in the section 3.2.1. A significant fraction (about 19%) of
BrC was associated with fossil fuel combustion. The fraction of secondary BrC was about
16.2%, which was enhanced during the summer due to the efficient photochemical formation
of secondary chromophores. The AAE value, closed to the aged SOA-AAE (4.7-5.3) (Bones
et al., 2010), can also verify it. The remaining fraction of BrC was derived from the fugitive
dust in the city. The results of BrC source apportionment for the Xi'an samples are in line
with the work by Shen et al. (2017) and also similar to the results obtained in Beijing by using
radiocarbon fingerprinting (Yan et al., 2017).
**4. Conclusions**

This study investigated the seasonality of the light-absorption characteristics of BrC in

Xi'an. Light absorption coefficient (MAC) of methanol-extracts at 365nm was 1.5-1.7 folds
higher than that of water-extracts in the two seasons, suggesting non-polar compounds in the
city are of stronger light-absorbing ability that that of polar compounds. The strong
correlation of levoglucosan with BrC and the diagnostic ratios of levoglucosan/mannosan and
levoglucosan/galacosan revealed that the wintertime abundant BrC ($abs_{\lambda=365nm}$-MeOH of
49.18 ± 31.67 M/m) in Xi'an was mainly derived from the residential biofuel combustion for
house heating in the city and its surrounding region. Size distribution results showed that 81%
and 65% of BrC occurred in the fine mode ($< 2.1\mu m$) during winter and summer, respectively,
which is characterized by a monomodal size distribution with a peak in winter and a bimodal
size distribution in summer with two peaks in the fine and coarse modes, respectively. The
fine mode of MAC is 50% higher than that in summer, suggesting that the light-absorbing
ability of wintertime fine particles is stronger, due to the abundant occurrence of PAHs and
other aromatic compounds in the fine mode. The linear correlation between the ratio of



$abs_{\lambda=365nm}$-MeOHO/$abs_{\lambda=550nm}$-EC and the enrichment of $^{13}$C during the haze development
indicated an accumulation of secondary BrC in the aerosol ageing process. The daytime
strong correlation of the ratio of $abs_{\lambda=365nm}$-MeOHO/$abs_{\lambda=550nm}$-EC with nitrophenols in the
haze event further revealed that such an enhanced production of secondary BrC is related to
the photooxidation of aromatic compounds with NOx. Source apportionment by using PMF
showed that 55% of the BrC was associated with biomass burning in the city during the
campaign, with 19 and 16% of BrC derived from fossil fuel combustion and secondary
formation, respectively.
Author contributions. GW designed the experiment. CW, JiaL, JinL and CC collected the
samples. CW and ZZ conducted the experiments. CW and GW performed the data
interpretation and wrote the paper. All authors contributed to the paper with useful scientific
discussions or comments.
Competing interests. The authors declare that they have no conflict of interest.

Acknowledgements. This work was financially supported by National Key R&D Plan
(Quantitative Relationship and Regulation Principle between Regional Oxidation Capacity of
Atmospheric and Air Quality (No. 2017YFC0210000), the program from National Nature
Science Foundation of China (No. 41773117).

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

## 733   Table List

Table 1. Concentrations of organic carbon in $PM_{2.5}$ and meteorological conditions during
winter and summer of 2017 in Xi'an, inland China.

Table 2. Comparison on light absorption ($abs_{\lambda=365nm}$), MAC, and AAE values of water-extracts of
$PM_{2.5}$ in Xi'an, China with those in other cities.

Table 3. Complex refractive index (k) of brown carbon from samples extracted by water in
two seasons.


## 744   Figure caption

Fig. 1 Temporal variations of WSOC, OC, $PM_{2.5}$, and $abs_{\lambda=365nm}$ of $PM_{2.5}$ samples extracted by
water ($H_2O$ extraction) and methanol (MeOH extraction) during winter (**a** and **c**) and summer
(**b** and **d**).

Fig. 2 Seasonal average values of $abs_{\lambda=365nm}$, AAE, and MAC extracted by MeOH and $H_2O$.
AAE is calculated by linear regression fit log ($abs_{\lambda=365nm}$) versus log($\lambda$) in the wavelength
range of 300–450 nm. (The shadows indicating the standard deviations)

Fig. 3 Size distributions of $abs_{\lambda=365nm}$ and MAC of $PM_{2.5}$ samples extracted by water during
the winter and summer of 2017 in Xi'an.

Fig. 4 Comparison of $abs_{\lambda=365nm}$ of samples between predicted by Mie theory and extracted by





water for different particle size (Dp < 2.1μm).

Fig. 4 Comparison of abs$_{\lambda=365nm}$ of samples predicted by Mie theorywith those of samples
extracted by water for different particle sizes (Dp < 2.1μm).

Fig. 5 Temporal variations of PM$_{2.5}$, meteorological parameters, abs$_{\lambda=365nm}$ of carbonaceous
matter and organic compounds in the period of January 10$^{th}$ -20$^{th}$ (The cyan shadow indicates
a haze period from January 12$^{th}$ to 19$^{th}$ with a daily PM$_{2.5}$ > 75 μg/m$^3$).

Fig.6 Linear fit regressions for the ratio of light absorption of methnoal-extracts to light
absorption of EC (abs$_{\lambda=365nm}$-MeOH/abs$_{\lambda=550nm}$-EC) with (**a** and **b**) δ$^{13}$C and (**c** and **d**) relative
abundance of nitrophenol to EC(Nitrophenol/EC) in the day- and night-PM$_{2.5}$ samples
collected during the haze period of January 12$^{th}$ to19$^{th}$ (corresponding to the cyan shadow in
Figure 5) in Xi'an.

Fig. 7 Factor profiles resolved by PMF mode during the winter and summer sampling period.
The bars represent the concentrations of species and the dots represent the contributions of
species appointed to the factors (the summer and winter samples were merged together for the
PMF analysis due to the limited number of samples).

Fig. 8 Source apportionment for airborne fine particulate BrC in Xi'an during the campaign.

Table 1. Concentrations of organic carbon in PM$_{2.5}$ and meteorological conditions during
winter and summer of 2017 in Xi'an, inland China.

|  | Winter | Summer |
|---|---|---|
| I. Mass concentrations of organic matter in PM$_{2.5}$ | | |
| WSOC (μg/m$^3$) | 23 ± 13 | 5.8 ± 1.4 |
| OC (μg/m$^3$) | 41 ± 25 | 8.4 ± 2.4 |
| PAHs (ng/m$^3$) | 149± 89 | 8.1 ± 6.5 |
| OPAHs (ng/m$^3$) | 174 ± 98 | 17 ± 8.7 |
| Nitrophenols (ng/m$^3$) | 17± 12 | 0.40 ± 0.27 |
| Levoglucosan (ng/m$^3$) | 739 ± 432 | 29 ± 22 |
| II. PM$_{2.5}$ and meteorological parameters | | |
| PM$_{2.5}$ (μg/m$^3$) | 194 ± 141 | 37 ± 16 |
| T (℃) | 2.6 ± 2.9 | 31 ± 5.4 |
| RH (%) | 60± 20 | 58 ± 19 |
| Visibility (km) | 7.0 ± 7.0 | 21 ± 11 |




Table 2. Comparison on light absorption ($abs_{\lambda=365nm}$), MAC, and AAE values of water-extracts of
PM$_{2.5}$ in Xi'an, China with those in other cities.

| Location | Time | abs$_{\lambda=365nm}$ (M/m) | | MAC (m$^2$/g) | | AAE | | References |
|---|---|---|---|---|---|---|---|---|
| | | Winter | Summer | Winter | Summer | Winter | Summer | |
| Xi'an, China | 2016-2017 | 49±32[a] | 5.2±2.1[a] | 1.3±0.03[a] | 0.8[a]±0.1[a] | 6.1±9.7[a] | 5.5±8.8[a] | This study |
| | | 28±16 | 3.5±1.7 | 1.2±0.06 | 1.1±0.2 | 5.3±8.5 | 4.8±7.7 | |
| | 2008-2009 | 46±20[a] | 8.3±2.3[a] | 1.3[a] | 0.7[a] | 6.0[a] | 6.0[a] | Huang et al. (2018) |
| | | 25±12 | 5.0±1.3 | 1.7 | 1.0 | 5.7 | 5.7 | |
| Beijing, China | 2010-2011 | 10±8.6 | 3.7±3.8 | 1.3 | 0.5 | | | Du et al. (2014) |
| | 2011 | 10±6.9 | | 1.2 | | 7.3 | | Cheng et al. (2016) |
| | 2013 | 14±5.2 | 4.6±2.2 | 1.5 | 0.7 | 5.3 | 5.8 | Yan et al. (2015) |
| Nanjing, China | 2015-2016 | 9.4 ± 4.7 | 3.3±2.4 | 1.0 | 0.5 | 6.7 | 7.3 | Chen et al. (2018) |
| Guangzhou, China | 2012 | 3.6±1.3 | | 0.8 | | 5.3 | | Liu et al. (2018) |
| Delhi, India | 2010-2011 | | | 1.6 | | 5.1 | | Kirillova et al. (2014) |
| Indo- Gangetic Plain India | 2015-2016 | 24±19 | | 1.2 | | | | Satish et al. (2017) |
| | 2011 | 40 ± 18[b] | | 1.3[b] | | 5.1[b] | | Bachi et al. (2016) |
| | | 52 ± 27[c] | | 1.3[c] | | 5.3[c] | | |
| Seoul, Korea | 2013-2013 | 11[a] | 5.8[a] | 0.9[a] | 1.5[a] | 5.5[a] | 4.1[a] | Kim et al.(2016) |
| | | 7.3 | 0.9 | 1.0 | 0.3 | 5.8 | 8.7 | |
| Atlanta, US | 2010 | | 0.6±0.4 | | 1.2-0.2 | | 3.4 | Zhang et al. (2011) |
| Los Angeles Basin, US | 2010 | | 0.4-1.6 | | 0.7 | | 7.6 | Zhang et al. (2013) |

Notes: [a] solution extracted by MeOH; [b] samples collected at day time; [c] samples collected in the night

Table 3. Complex refractive index (k) of brown carbon from samples extracted by water in
two seasons.

| Particle size (μm) | Winter | Summer |
|---|---|---|
| 1.31 | 0.047 ± 0.005 | 0.021 ± 0.010 |
| 0.73 | 0.048 ± 0.008 | 0.033 ± 0.010 |
| 0.45 | 0.048 ± 0.013 | 0.031 ± 0.009 |
| 0.18 | 0.038 ± 0.016 | 0.026 ± 0.008 |


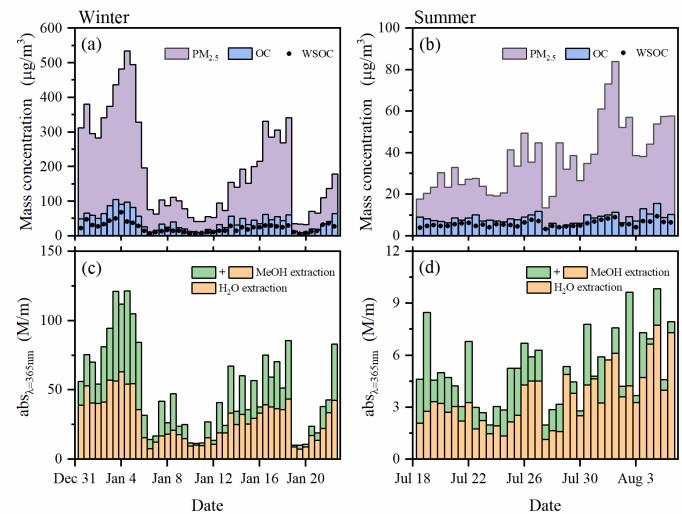


Fig. 1 Temporal variations of WSOC, OC, PM$_{2.5}$, and abs$_{\lambda=365nm}$ of PM$_{2.5}$ samples extracted by
water (H$_2$O-extraction) and methanol (MeOH-extraction) during winter (**a** and **c**) and summer
(**b** and **d**).


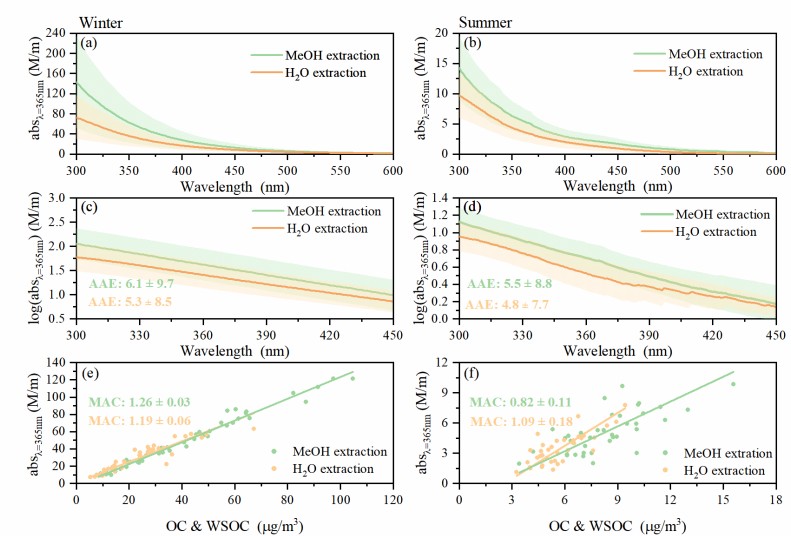



Fig. 2 Seasonal average values of abs$_{\lambda=365nm}$, AAE, and MAC extracted by MeOH and H$_2$O.
AAE is calculated by linear regression fit log (abs$_{\lambda=365nm}$) versus log($\lambda$) in the wavelength
range of 300–450 nm. (The shadows indicating the standard deviations)



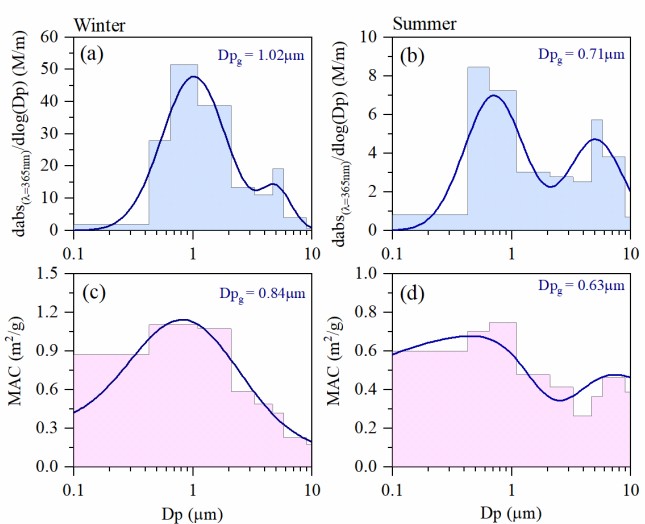

Fig. 3 Size distributions of $abs_{\lambda=365nm}$ and MAC of $PM_{2.5}$ samples extracted by water during the winter and summer of 2017 in Xi'an.

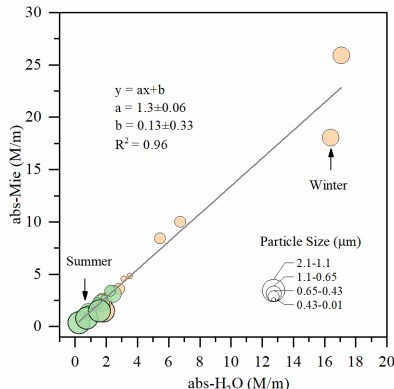

Fig. 4 Comparison of $abs_{\lambda=365nm}$ of samples predicted by Mie theory with those of samples extracted by water for different particle sizes ($Dp < 2.1\mu m$).


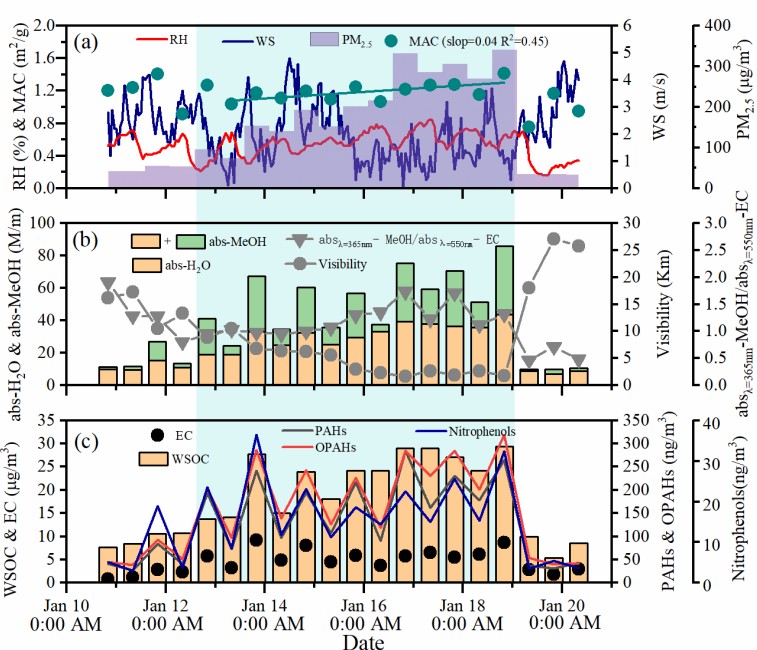


Fig. 5 Temporal variations of PM$_{2.5}$, meteorological parameters, abs$_{\lambda=365nm}$ of carbonaceous matter and organic compounds in the period of January 10$^{th}$ -20$^{th}$ (The cyan shadow indicates a haze period from January 12$^{th}$ to 19$^{th}$ with a daily PM$_{2.5}$ > 75 µg/m$^3$).





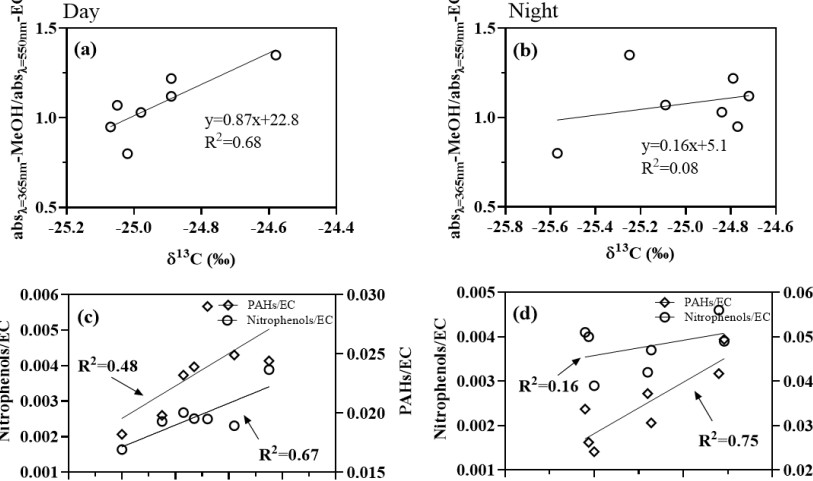

Fig.6 Linear fit regressions for the ratio of light absorption of methnoal-extracts to light
absorption of EC (abs$_{\lambda=365nm}$-MeOH/abs$_{\lambda=550nm}$-EC) with (**a** and **b**) $\delta^{13}$C and (**c** and **d**) relative
abundance of nitrophenol to EC(Nitrophenol/EC) in the day- and night-PM$_{2.5}$ samples
collected during the haze period of January 12$^{th}$ to19$^{th}$ (corresponding to the cyan shadow in
Figure 5) in Xi'an.

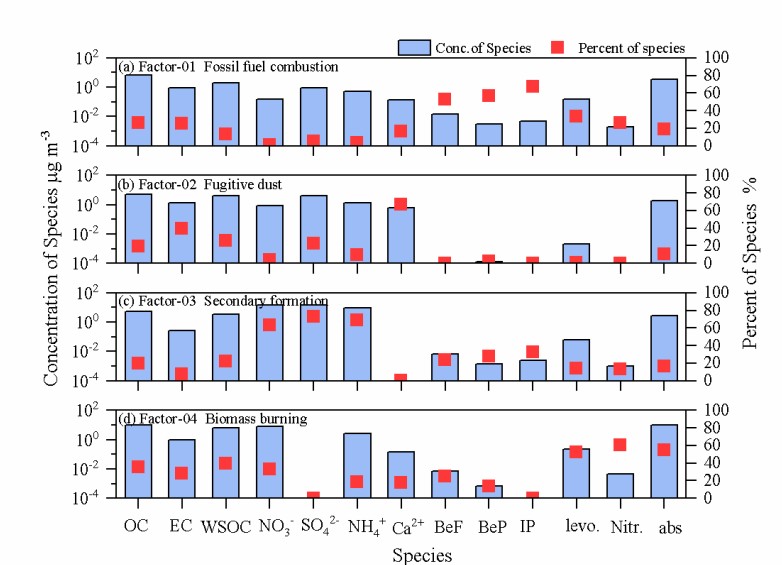


Fig. 7 Factor profiles resolved by PMF mode during the winter and summer sampling period.
The bars represent the concentrations of species and the dots represent the contributions of
species appointed to the factors (the summer and winter samples were merged together for the
PMF analysis due to the limited number of samples).






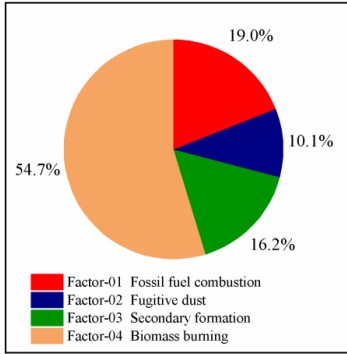


Fig. 8 Source apportionment for airborne fine particulate BrC in Xi'an during the campaign.