# Peer review of "The characteristics of atmospheric brown carbon in Xi'an, inland China: sources, size distributions and optical properties Can Wu1,2, Gehui Wang1,2,3,4\*, Jin Li2, Jianjun Li2, Cong Cao2, Shuangshuang Ge1, Yuning Xie1,"

_Atmospheric Chemistry and Physics, 2019_

## Referee Comment (RC1) · Anonymous Referee #2 · 17 Sep 2019

This paper reports on the light absorption of organic aerosol components (BrC) in an urban setting with significant biomass burning emissions in China. Both the experimental methods and the data analyses approaches are largely identical to other studies and are not novel. This paper's contribution is that it adds more data points to the characterization of BrC in various locations. Possibly the most interesting finding is strong evidence for the secondary formation of BrC in summer. Overall I have only minor comments, with the exception that 1) the authors could do a better job of citing original sources, and 2) the paper could use some editing throughout. The paper is appropriate for ACP and in my view acceptable after consideration of the following issues.

[Figure]

Specific comments

Line 160 to 161. In calculating the MAC what is the concentration of the the carbon? That is, clarify what is the WSOC and MSOC concentration. I assume they are mass of carbon per volume of air, not per volume of liquid extract. Also, what about conversion of carbon mass to organic aerosol mass. Finally, it seems that since this is all liquid based analysis, can this data be applied to atmospheric aerosols? If the authors think so, justify how this is done (see the comment below on the 1.3 factor, this should be noted here in the text).

It would clarify things if the authors used the units of ugC/m3 for WSOC and OC throughout, instead of ug/m3.

Line 198 and beyond. The use of the term matters is a poor choice. Eg, edit . . . as ubiquitous matters in the atmosphere. Also in a few lines down the term matters is again used.

Edit line 201, due to owning. . .?

Line 314 to 317. The value of Mei-predicted (based on size resolved data) and bulk light absorption factor of 1.3 is interesting. Since this work is simply repeating what other studies have done, please compare this 1.3 factor to the factor reported in these other studies (these papers are already cited). Also, the statement that using liquid based Abs or MACs to estimate aerosol optical effects will result in an underestimation is obvious and not new. It seems more important that the authors should state that in this study, to convert the liquid based data reported here to estimated aerosol properties, a factor of 1.3 must be applied, at least for the water-soluble data.

In Figure 5, explain the regular up and down pattern in PAHs, OPAHs and nitrophenols. The pattern also seems to be evident in some other plotted factors, such as Abs. . . This looks like a measurement artifact and not real. Please provide some evidence it is real or possible explanation for the cause.

Line 346, or at least the bleaching was offset by SOA BrC formation.

Line 350, why is the term mass absorption efficiency used here where earlier the MAC was discussed. What is the difference?

I suggest throughout the paper the authors do not give % fraction of the various sources to one decimal place, the precision is not that high (ie, convert 16.2% to 16%, etc).

Edit line 411-412

---

## Referee Comment (RC2) · Anonymous Referee #3 · 30 Nov 2019

This study characterized brown carbon including extracted mass using methanol and water, and PAHs, Oxy-PAHs etc., composition analysis for size-segregated particles collected during winter and summer period in Xi'an. Brown carbon has significant impacts on radiative forcing and regional climate, and receives growing interests during the past decades. Studies on abundance, temporal variations, possible sources, ect., of BrC is essential for a better understanding on impacts of carbonaceous aerosols in environment on air quality and climate, and vital for air pollution controls. My main questions on this study are:

Line 46, for correlation analysis, use r instead of R2. Please correct and revise through-

out the manuscript

Line 56, pay more attention to the significant figures

Line 63, "a small fraction"- do you mean the mass fraction? Giving high OC fractions in carbonaceous aerosols, and probably high abundance of BrC to total OC, its mass fraction might be not "small", although so far it is still unclear about its mass as there is only an operationally defined term, and mass differs when using different extraction methods.

Lines 102-105, better to move this part ahead to the beginning of paragraph

Line 112, "12-hr" – only daytime, or both day and night samples were collected?

Line 160, provide methods in quantification mass of WSOC and MSOC

Line 172, why not "300-880nm"?

Line 195, start a new sentence "similar phenomena. . ."

Line 198 "mainly"- the relative contribution of primary and secondary sources for OPAHs and nitrophenols is still unclear so far. But my opinion here is that the word "mainly" here might be inappropriate. Also, the authors may provide some past emission studies on OPAHs, nitrophenols, and PAHs here. The cited study here was insufficient to support the statement.

Lines 201-203, but the question is that these chemicals comprised only a small mass fraction of BrC or OC, so the light absorption of BrC could be attributed to other components, although currently there is a big gap in this area.

Line 205, a few past studies indicated that in some area especially in north China, coal burning could be an unignorable source of LG as well.

Line 216, "open biomass burning"

Line 233, is it possible that BrC compositions differed among these sites, resulting

different absorption efficient? "BrC pollution is more significnat"- is a little hard to understand.

Line 237, "OC" or "MSOC"?

Line 254- any evidence or past studies to support this?

Lines 293-317, while interpreting these results, I'd like to suggest to paying more attentions to the uncertainties in both Mie theory calculation as well as experimental methods, and difference in difference extraction approaches. The 30% difference may be not a "significant underestimation".

Line 347, delete "which is also called black carbon"

Line 411, did the authors calculate AAE for this fraction(source) separately? Please clarify.

Figure 6- suggest to improving quality

---

## Author Comment (AC1) · 25 Dec 2019

Dear Atmospheric Chemistry and Physics Editor:

After reading the comments from the reviewer, we have carefully revised our manuscript. Our responses to the comments are itemized below.

Anything for our paper, please feel free to contact me via cwu@geo.ecnu.edu.cn, or ghwang@geo.ecnu.edu.cn.

All the best

Can Wu
On behalf of Prof. Gehui Wang
December 25, 2019

Reviewer(s)' Comments to Author:

**Reviewer 2**
**Comments:**
This paper reports on the light absorption of organic aerosol components (BrC) in an urban setting with significant biomass burning emissions in China. Both the experimental methods and the data analyses approaches are largely identical to other studies and are not novel. This paper's contribution is that it adds more data points to the characterization of BrC in various locations. Possibly the most interesting finding is strong evidence for the secondary formation of BrC in summer. Overall, I have only minor comments, with the exception that 1) the authors could do a better job of citing original sources, and 2) the paper could use some editing throughout. The paper is appropriate for ACP and in my view acceptable after consideration of the following issues.
**Reply**: We thank the reviewer's valuable comments. We have carefully revised our manuscript according to your advice. See details below.

**Comments:**
1) Line 160 to 161. In calculating the MAC what is the concentration of the carbon? That is, clarify what is the WSOC and MSOC concentration. I assume they are mass of carbon per volume of air, not per volume of liquid extract. Also, what about conversion of carbon mass to organic aerosol mass. Finally, it seems that since this is all liquid based analysis, can this data be applied to atmospheric aerosols? If the authors think so, justify how this is done (see the comment below on the 1.3 factor, this should be noted here in the text).
**Reply**: As for calculating the MAC values, the WSOC and OC are the mass concentrations of carbon per volume of air ($ugC/m^3$), respectively. We didn't convert the carbon mass to

organic aerosol mass, because it would not give any extra information and numerous studies were processed in a similar way (Liu et al., 2013;Kirillova et al., 2014). Although the liquid extraction method may underestimate the light-absorption of brown carbon, it is an extensive research technique for BrC at present with many advantages, e.g., without interference of black carbon, wavelength continuity, and so on. To a certain extent, the results can explain the BrC light-absorption, and it may correct based on the data by Mie theory predicted to better quantify the BrC absorption in the future studies.

**Comments:**
2) It would clarify things if the authors used the units of ugC/m$^3$ for WSOC and OC throughout, instead of ug/m$^3$.
**Reply**: Suggestion taken. We have replaced the unit for WSOC and OC throughout the revised manuscript.

**Comments:**
3) Line 198 and beyond. The use of the term matters is a poor choice. Eg, edit : as ubiquitous matters in the atmosphere. Also in a few lines down the term matters is again used.
**Reply**: Thank for the comments, we changed the "matters" into "compounds,".

**Comments:**
4) Edit line 201, due to owning: : :?
**Reply**: Suggestion taken. We have modified it in the revised manuscript.

**Comments:**
5) Line 314 to 317. The value of Mei-predicted (based on size resolved data) and bulk light absorption factor of 1.3 is interesting. Since this work is simply repeating what other studies have done, please compare this 1.3 factor to the factor reported in these other studies (these papers are already cited). Also, the statement that using liquid based Abs or MACs to estimate aerosol optical effects will result in an underestimation is obvious and not new. It seems more important that the authors should state that in this study, to convert the liquid based data reported here to estimated aerosol properties, a factor of 1.3 must be applied, at least for the water-soluble data.
**Reply**: Suggestion taken, we had one sentence to state this issue, see page 15, line 318-321.

**Comments:**
6) In Figure 5, explain the regular up and down pattern in PAHs, OPAHs and nitrophenols. The pattern also seems to be evident in some other plotted factors, such as Abs: : : This looks like a measurement artifact and not real. Please provide some evidence it is real or possible explanation for the cause.
**Reply**: The regular up and down pattern in PAHs and related compounds is real, which was mainly caused by the lower PBL height and higher emissions at night; the high concentrations were for the nighttime samples and the low concentrations were for the

daytime samples. Here we give more evidences to show that the variation pattern is real. As seen in Fig. 1(a), PAHs showed a robust linear correlation with EC, suggesting that the pattern of PAHs should be similar with EC (Figure 1(b)). This can be interpreted by the change of boundary layer and intense biomass burning at nigh, which reveals a real variation tendency of PAHs during the sampling period. Moreover, as shown in Figure 1c, the triplicate results of the recovery experiments for the target compounds (i.e., PAHs) are very stable with a relative standard deviation (RSD) less than 5%, further demonstrating the data reported here are real. Therefore, we believe that our results presented by this work are accurate and reasonable.

[Figure]

Figure 1. (a) The relationship of EC and PAHs during the haze period of January 12th to19th. (b) Temporal variations of EC and PAHs during the haze period of January 12th to19th. (c) The data for standard PAHs experiments of repeatability and recovery. (RSD: relative standard deviation, the concentration of standards is 0.8 ng/μl)

**References**

Kirillova, E. N., Andersson, A., Tiwari, S., Srivastava, A. K., Bisht, D. S., and Gustafsson, Ö.: Water-soluble organic carbon aerosols during a full New Delhi winter: Isotope-based source apportionment and optical properties, Journal of Geophysical Research: Atmospheres, 119, 3476-3485, 10.1002/2013jd020041, 2014.

Liu, J., Bergin, M., Guo, H., King, L., Kotra, N., Edgerton, E., and Weber, R. J.: Size-resolved measurements of brown carbon in water and methanol extracts and estimates of their contribution to ambient fine-particle light absorption, Atmospheric Chemistry and Physics, 13, 12389-12404, 10.5194/acp-13-12389-2013, 2013.

---

## Author Comment (AC2) · 25 Dec 2019

Dear Atmospheric Chemistry and Physics Editor:

After reading the comments from the reviewer, we have carefully revised our manuscript. Our responses to the comments are itemized below.

Anything for our paper, please feel free to contact me via cwu@geo.ecnu.edu.cn, or ghwang@geo.ecnu.edu.cn.

All the best

Can Wu
On behalf of Prof. Gehui Wang
December 25, 2019

Reviewer(s)' Comments to Author:

**Reviewer 3**
**Comments:**
This study characterized brown carbon including extracted mass using methanol and water, and PAHs, Oxy-PAHs etc., composition analysis for size-segregated particles collected during winter and summer period in Xi'an. Brown carbon has significant impacts on radiative forcing and regional climate, and receives growing interests during the past decades. Studies on abundance, temporal variations, possible sources, ect., of BrC is essential for a better understanding on impacts of carbonaceous aerosols in environment on air quality and climate, and vital for air pollution controls.
**Reply**: We thank the reviewer's valuable comments. We have carefully revised our manuscript according to her/his advice. See details below.

**Comments:**
1) Line 46, for correlation analysis, use r instead of $R^2$. Please correct and revise throughout the manuscript.
**Reply**: Suggestion taken. We have substituted r for $R^2$ in both revised draft and supporting information.

**Comments:**
2) Line 56, pay more attention to the significant figures.
**Reply**: Suggestion taken. We revised the format. (Please see line 56)

**Comments:**
3) Line 63, "a small fraction"- do you mean the mass fraction? Giving high OC fractions

in carbonaceous aerosols, and probably high abundance of BrC to total OC, its mass fraction might be not "small", although so far it is still unclear about its mass as there is only an operationally defined term, and mass differs when using different extraction methods.
**Reply**: We do agree with the comments above, and revised the statement. See page 3, line 63-64.

**Comments:**
4) Lines 102-105, better to move this part ahead to the beginning of paragraph.
**Reply**: Suggestion taken. See page 4, line 98-101.

**Comments:**
5) Line 112, "12-hr" – only daytime, or both day and night samples were collected?
**Reply**: The samples were collected on a day/night basis with each for 12-hrs. We have rephrased the descriptions. See page 5, line 112-114.

**Comments:**
6) Line 160, provide methods in quantification mass of WSOC and MSOC.
**Reply**: Suggestion taken. As for the WSOC measurement, we added the related information into the experiment section, see page 6, line 129-132. Because MSOC cannot be measured directly, we use OC to replace it for $BrC_{-methonal}$ calculation, which is a common way in aerosol community for BrC study. The related explanation was also given in the revised version, see page 7, line 159-163.

**Comments:**
7) Line 172, why not "300-880nm"?
**Reply**: The absorptions for aerosol extracts were almost zero at the wavelengths above 700 nm, hence, the AAE calculation by previous studies usually selected different wavelength ranges, e.g., 300 -500 nm (Liu et al., 2013), 310 -450 nm (Cheng et al., 2016), 330- 400 nm (Kirillova et al., 2014). In fact, we recalculated the AAE values at 300-500 nm for both extractions in winter, but we found that the variance for both wavelength ranges was little (< 10%), and have minor effect for our conclusion. Therefore, we believe the current wavelengths we selected are reasonable.

**Comments:**
8) Line 195, start a new sentence "similar phenomena: : :"
**Reply**: Suggestion taken. See page 9, line 192-193.

**Comments:**
9) Line 198 "mainly"- the relative contribution of primary and secondary sources for OPAHs and nitrophenols is still unclear so far. But my opinion here is that the word "mainly" here might be inappropriate. Also, the authors may provide some past emission studies on OPAHs, nitrophenols, and PAHs here. The cited study here was insufficient to support the statement.

**Reply**: Suggestion taken. We have revised the discussions. See page 9, line 195-200.

**Comments:**
10) Lines 201-203, but the question is that these chemicals comprised only a small mass fraction of BrC or OC, so the light absorption of BrC could be attributed to other components, although currently there is a big gap in this area.
**Reply**: We agree with the reviewer that these compounds comprised only a small fraction of BrC, and the light absorption of BrC could be attributed to other components. As mentioned by the reviewer, currently only a small fraction of BrC can be characterized on a molecular level and there is a big gap in this area. Many studies have found that PAHs, OPAHs and nitrophenols are of strong light-absorbing ability, although their contributions to the total light-absorption are relatively small mainly due to their very low mass fractions. However, their sources especially secondary formation pathways are still not clear. Therefore, studies related to these BrC compounds are warranted.

**Comments:**
11) Line 205, a few past studies indicated that in some area especially in north China, coal burning could be an unignorable source of LG as well.
**Reply**: We agree that coal burning is an unignorable source of levoglucosan in some area of China. However, we found that the ratio of the levoglucosan/mannosan and levoglucosan/galacosan in Xi'an during the campaign are similar to the biomass types (see in Fig. S2). Therefore, we believe that in Xi'an and its surrounding regions levoglucosan is mostly derived from biomass burning and can be taken as the tracer. The related discussion can be found in page 11, line 243-247, and the supporting information (Figure S2).

**Comments:**
12) Line 216, "open biomass burning"
**Reply**: Suggestion taken.

**Comments:**
13) Line 233, is it possible that BrC compositions differed among these sites, resulting different absorption efficient? "BrC pollution is more significant"- is a little hard to understand.
**Reply**: Thank for your valuable advice. We think the higher values of $abs_{\lambda=365nm}$ values in Xi'an than in Beijing, US and Korea are mainly due to the higher level of light-absorbing compounds in the urban atmosphere. We agree with the reviewer that BrC compositions differed among these sites, resulting different absorption efficiency. As defined in page 7, MAC is $abs_{\lambda=365nm}$ divided by WSOC or OC, which reflecting the light-absorbing ability of per unit mass of BrC. From Table 2, we can see that MAC in Xi'an during the two seasons are higher than those in US and Korea, suggesting that BrC in Xi'an is comprised of stronger light-absorbing compounds. We added this discussion into the text, see page, line 239-241. We also modified the sentence "BrC pollution is more significant" as suggesting a heavy pollution of light-absorbing aerosols in Xi'an. See page10, line 231.

**Comments:**

14) Line 237, "OC" or "MSOC"?

**Reply**: We assumed that OC could be completely dissolved in methanol solvent and substituted MSOC, so it is WSOC or MSOC here. We changed the OC term into MSOC, to keep consistent. See page 11, line 235.

**Comments:**

15) Line 254- any evidence or past studies to support this?

**Reply**: Phthalates, as important plasticizer, are easily dissolved in methanol, but not in water. It has less absorption at wavelength >300 nm (Du et al., 2014). We analyzed 3 species of phthalates, and found that phthalates/OC ratio was about 10 times higher in summer than in winter (Figure 1). Hence, we think that more amount of non-BrC by methanol extraction lead to smaller $MAC_{MSOC}$ compared with $MAC_{WSOC}$ in summer. We added this explanation into the text. See page 12, line255-266.

[Figure]

Figure 1 The ratio of phthalates/OC in both seasons.

**Comments:**

16) Lines 293-317, while interpreting these results, I'd like to suggest to paying more attentions to the uncertainties in both Mie theory calculation as well as experimental methods, and difference in difference extraction approaches. The 30% difference may be not a "significant underestimation".

**Reply**: Thank you for reminding. An orthogonal regression was applied here, which was better than the previous linear fit due to considering the errors with variables x and y. From Figure 2a, the slope has changed compared with previous method (1.3), and the variance of variable x is significant. We noted that one point, as depicted in Figure 2a (red circle), is much far away from the fitting line. Hence, it may be an abnormal datum, and thus lead to this phenomenon. Unfortunately, there are no more filter for measuring this sample again. So, we have to exclude this point, and found that the factor become bigger than that depicted in Figure 2a. While, the variance of variable x (0.04) is only 0.1fold for the previous one. To make the data more representative, we also choose the $5^{th} \sim 95^{th}$ percentiles of all data points to further verify the result, which is same as that in Fig. 2b. To sum up, the factor is about 1.5 between abs-Mie and abs-measure rather than 1.3. Therefore, our previous conclusion about underestimation of BrC absorption by solvent extraction methods is reasonable. We recommend a factor of 1.5 to o convert the liquid-based data (at least for the water-soluble data) reported by this work for estimating optical properties of atmospheric aerosols in Xi'an and its surrounding regions in order to

better quantify the BrC light-absorption. We added this discussion into the text, See page 15, line 318-321.

[Figure]

Figure 2 An orthogonal regression for different data. (a) all data points, (b) excepting abnormal point (red circle in Fig. 2(a)), (c) 5th ~ 95th percentiles of all data points.

**Comments:**
17) Line 347, delete "which is also called black carbon"
**Reply**: Suggestion taken.

**Comments:**
18) Line 411, did the authors calculate AAE for this fraction(source) separately? Please clarify.
**Reply**: In this study, we have no way to calculate the AAE values for each source, so the value represents the AAE of total BrC. We have clarified this issue in the revised manuscript. See page 19, line 414.

**Comments:**
19) Figure 6- suggest to improving quality
**Reply**: Suggestion taken. We have replotted it.

**References**

Cheng, Y., He, K.-b., Du, Z.-y., Engling, G., Liu, J.-m., Ma, Y.-l., Zheng, M., and Weber, R. J.: The characteristics of brown carbon aerosol during winter in Beijing, Atmospheric Environment, 127, 355-364, 10.1016/j.atmosenv.2015.12.035, 2016.

Cochran, R. E., Jeong, H., Haddadi, S., Fisseha Derseh, R., Gowan, A., Beránek, J., and Kubátová, A.: Identification of products formed during the heterogeneous nitration and ozonation of polycyclic aromatic hydrocarbons, Atmospheric Environment, 128, 92-103, 10.1016/j.atmosenv.2015.12.036, 2016.

Du, J.-B., Tang, Y.-L., Long, Z.-W., Hu, S.-H., and Li, T.: Theoretical calculation of spectra of dibutyl phthalate and dioctyl phthalate, Russian Journal of Physical Chemistry A, 88, 819-822, 10.1134/s0036024414050100, 2014.

Huang, R. J., Yang, L., Cao, J., Chen, Y., Chen, Q., Li, Y., Duan, J., Zhu, C., Dai, W., and Wang, K.: Brown Carbon Aerosol in Urban Xi'an, Northwest China: The Composition and Light Absorption Properties, Environmental Science & Technology, 52, 6825-6833 10.1021/acs.est.8b02386, 2018.

Keyte, I. J., Harrison, R. M., and Lammel, G.: Chemical reactivity and long-range transport potential of polycyclic aromatic hydrocarbons - a review, Chemical Society Reviews, 42, 9333-9391, 10.1039/c3cs60147a, 2013.

Kirillova, E. N., Andersson, A., Tiwari, S., Srivastava, A. K., Bisht, D. S., and Gustafsson, Ö.: Water-soluble organic carbon aerosols during a full New Delhi winter: Isotope-based source apportionment and optical properties, Journal of Geophysical Research: Atmospheres, 119, 3476-3485, 10.1002/2013jd020041, 2014.

Liu, J., Bergin, M., Guo, H., King, L., Kotra, N., Edgerton, E., and Weber, R. J.: Size-resolved measurements of brown carbon in water and methanol extracts and estimates of their contribution to ambient fine-particle light absorption, Atmospheric Chemistry and Physics, 13, 12389-12404, 10.5194/acp-13-12389-2013, 2013.

Shen, G., Tao, S., Wei, S., Chen, Y., Zhang, Y., Shen, H., Huang, Y., Zhu, D., Yuan, C., Wang, H., Wang, Y., Pei, L., Liao, Y., Duan, Y., Wang, B., Wang, R., Lv, Y., Li, W., Wang, X., and Zheng, X.: Field measurement of emission factors of PM, EC, OC, parent, nitro-, and oxy- polycyclic aromatic hydrocarbons for residential briquette, coal cake, and wood in rural Shanxi, China, Environ Sci Technol, 47, 2998-3005, 10.1021/es304599g, 2013.

Wang, G., Xie, M., Hu, S., Gao, S., Tachibana, E., and Kawamura, K.: Dicarboxylic acids, metals and isotopic compositions of C and N in atmospheric aerosols from inland China: implications for dust and coal burning emission and secondary aerosol formation, Atmospheric Chemistry and Physics, 10, 6087-6096, 10.5194/acp-10-6087-2010, 2010.

Wang, G. H., Zhou, B. H., Cheng, C. L., Cao, J. J., Li, J. J., Meng, J. J., Tao, J., Zhang, R. J., and Fu, P. Q.: Impact of Gobi desert dust on aerosol chemistry of Xi'an, inland China during spring 2009: differences in composition and size distribution between the urban ground surface and the mountain atmosphere, Atmospheric Chemistry and Physics, 13, 819-835, 10.5194/acp-13-819-2013, 2013.

Yuan, B., Liggio, J., Wentzell, J., Li, S.-M., Stark, H., Roberts, J. M., Gilman, J., Lerner, B., Warneke, C., Li, R., Leithead, A., Osthoff, H. D., Wild, R., Brown, S. S., and de Gouw, J. A.: Secondary formation of nitrated phenols: insights from observations during the Uintah BasinWinter Ozone Study (UBWOS) 2014, Atmospheric Chemistry and Physics, 16, 2139-2153, 10.5194/acp-16-2139-2016, 2016.

Zhang, Y., and Tao, S.: Global atmospheric emission inventory of polycyclic aromatic hydrocarbons (PAHs) for 2004, Atmospheric Environment, 43, 812-819, 10.1016/j.atmosenv.2008.10.050, 2009.